# Association of Gulf War Illness with Characteristics in Deployed vs. Non-Deployed Gulf War Era Veterans in the Cooperative Studies Program 2006/Million Veteran Program 029 Cohort: A Cross-Sectional Analysis

**DOI:** 10.3390/ijerph20010258

**Published:** 2022-12-24

**Authors:** Linh M. Duong, Alice B. S. Nono Djotsa, Jacqueline Vahey, Lea Steele, Rachel Quaden, Kelly M. Harrington, Sarah T. Ahmed, Renato Polimanti, Elani Streja, John Michael Gaziano, John Concato, Hongyu Zhao, Krishnan Radhakrishnan, Elizabeth R. Hauser, Drew A. Helmer, Mihaela Aslan, Elizabeth J. Gifford

**Affiliations:** 1Cooperative Studies Program Clinical Epidemiology Research Center (CSP-CERC), VA Connecticut Healthcare System 151B, West Haven, CT 06516, USA; 2Department of Psychiatry, Yale School of Medicine, Yale University, New Haven, CT 06511, USA; 3Center for Innovations in Quality, Effectiveness, and Safety (IQuESt), Michael E. DeBakey VA Medical Center, Houston, TX 77030, USA; 4Department of Medicine, Baylor College of Medicine, Houston, TX 77030, USA; 5VA Cooperative Studies Program Epidemiology Center—Durham, Department of Veterans Affairs, Durham, NC 27705, USA; 6Computational Biology and Bioinformatics Program, Duke University, Durham, NC 27705, USA; 7Veterans Health Research Program, Yudofsky Division of Neuropsychiatry, Department of Psychiatry and Behavioral Sciences, Baylor College of Medicine, Houston, TX 77030, USA; 8Massachusetts Veterans Epidemiology Research and Information Center (MAVERIC), VA Boston Healthcare System, Boston, MA 02130, USA; 9Department of Psychiatry, Boston University School of Medicine, Boston, MA 02118, USA; 10Department of Medicine, Brigham and Women’s Hospital, Harvard Medical School, Boston, MA 02115, USA; 11Yale School of Medicine, Yale University, New Haven, CT 06511, USA; 12Food and Drug Administration, Silver Spring, MD 20993, USA; 13Department of Biostatistics, Yale School of Public Health, New Haven, CT 06520, USA; 14National Mental Health and Substance Use Policy Laboratory, Substance Abuse and Mental Health Services Administration, Rockville, MD 20857, USA; 15Department of Biostatistics and Bioinformatics, Duke Molecular Physiology Institute, Duke University, Durham, NC 27705, USA; 16Center for Child and Family Policy, Duke Margolis Center for Health Policy, Duke University Sanford School of Public Policy, Durham, NC 27708, USA

**Keywords:** gulf war, gulf war illness, chronic multisymptom illness, post-deployment health surveys, health outcomes, veteran

## Abstract

Gulf War Illness (GWI), a chronic multisymptom illness with a complex and uncertain etiology and pathophysiology, is highly prevalent among veterans deployed to the 1990–1991 GW. We examined how GWI phenotypes varied by demographic and military characteristics among GW-era veterans. Data were from the VA’s Cooperative Studies Program 2006/Million Veteran Program (MVP) 029 cohort, Genomics of GWI. From June 2018 to March 2019, 109,976 MVP enrollees (out of a total of over 676,000) were contacted to participate in the 1990–1991 GW-era Survey. Of 109,976 eligible participants, 45,169 (41.1%) responded to the 2018–2019 survey, 35,902 respondents met study inclusion criteria, 13,107 deployed to the GW theater. GWI phenotypes were derived from Kansas (KS) and Centers for Disease Control and Prevention (CDC) GWI definitions: (a) KS Symptoms (KS Sym+), (b) KS GWI (met symptom criteria and without exclusionary health conditions) [KS GWI: Sym+/Dx−], (c) CDC GWI and (d) CDC GWI Severe. The prevalence of each phenotype was 67.1% KS Sym+, 21.5% KS Sym+/Dx−, 81.1% CDC GWI, and 18.6% CDC GWI severe. These findings affirm the persistent presence of GWI among GW veterans providing a foundation for further exploration of biological and environmental underpinnings of this condition.

## 1. Introduction

Gulf War Illness (GWI) is a complex, chronic multisymptom condition with uncertain etiology and pathophysiology among GW-era veterans who deployed in support of Operations Desert Shield and Desert Storm [1,2,3,4,5,6]. GW-era veterans may have been exposed to a variety of environmental and chemical hazards that may pose potential health risks [3]. Clinically, veterans with GWI present with chronic symptoms such as fatigue, pain, gastrointestinal symptoms, respiratory symptoms, dermatological symptoms, and neurological symptoms that are not better explained by other diagnoses [1,2,3,4,5,6,7,8]. Decades since the GW ended, GW veterans continued to report poorer health related to symptoms and multiple co-morbid conditions [4,9]. While higher rates of GWI has been documented among deployed GW veterans, GWI has also been observed among non-deployed to a lesser extent [10,11]

There is no diagnostic biomarker or objective laboratory test with which to confirm a GWI case [12]. Instead, GWI is identified by patient self-report of the presence and severity of specific symptoms and the absence of exclusionary health conditions that may explain these symptoms [7,8,13,14,15,16,17]. For research purposes, the 2014 U.S. Institute of Medicine (IOM) report [14] recommended the use of two GWI case definitions developed by Steele (Kansas definition) [8] and Fukuda et al. (Centers for Disease Control and Prevention [CDC] definition) [7] due to their close alignment with multisystem patterns of symptoms reported among deployed GW veterans. While both definitions are based on symptoms reported by veterans in multiple domains (detailed in the methods section), key differences exist. The Kansas GWI definition considers the presence of certain health conditions as a disqualification for consideration of GWI. Thus, veterans with certain health conditions are considered non-cases. In contrast, the CDC GWI definition does not consider health conditions as exclusionary conditions. Further, the two definitions differ in the number of symptoms and symptom domains as well as how the number and severity of symptoms within each domain are considered. Careful consideration of such details in how a GWI phenotype is assigned is essential for constructing GWI phenotypes that serve as the basis for future studies to examine associations with factors such as exposures, genetic variation, and biophysiological changes.

The Department of Veterans Affairs’ (VA) Million Veteran Program (MVP), the nation’s largest repository of veteran data and biospecimens, is one of the most diverse cohorts in the world [18]. MVP’s overarching goal is to utilize research findings to refine and deliver personalized medicine of relevance to veterans [18]. The current study seeks to contribute to the GWI literature by leveraging VA research infrastructure to study a large sample of GW-era veterans who have provided information regarding current symptoms, diagnoses, and characteristics of their military service. This study addressed four aims: (1) to describe demographic and military service differences between deployed and non-deployed GW-era veteran cohorts, (2) to report the prevalence among GW-era MVP participants of GWI phenotypes as defined by Kansas and CDC definitions stratified by deployment status and to assess associations with deployment; (3) to evaluate and compare exclusionary conditions by deployment status; and (4) to assess and compare associations among sociodemographic and military characteristics and GWI phenotypes among deployed and non-deployed GW-era veterans. These findings will form the foundation for additional epidemiologic, clinical, and genetic analyses of this cohort.

## 2. Materials and Methods

### 2.1. Data & Participants

MVP is an observational cohort study and mega-biobank within the VA healthcare system, combining the VA electronic health records (EHR) system with genetic data [19]. The VA’s Cooperative Studies Program 2006/Million Veteran Program 029 cohort, Genomics of GWI (VA CSP 2006/MVP 029 Project) [20], was established to investigate the genetic and military exposure underpinnings of GWI in a cohort of 1990–1991 GW-era veterans enrolled in MVP. From June 2018 to March 2019, 109,976 MVP enrollees (out of a total of over 676,000) were contacted to participate in the 1990–1991 GW-era Survey. These veterans met the following criteria: 1. Information in VA/Department of Defense Identity Repository (VADIR) indicated military service between 2 August 1990 and 31 July 1991 including Active Duty, Reserves, or National Guard service; 2. MVP blood specimen was available; 3. Participant was not deceased, 4. Had not withdrawn from MVP; 5. Did not opt-out of receiving additional research requests; 6. Was not included in a previous pilot mailing to GW-era veterans; and 7. Had an accurate and current address on file (Kelly Harrington, Department of Veterans Affairs, “Personal communication”, 2021). Veterans meeting these criteria were sent a questionnaire to collect information on symptoms, diagnoses, military service, deployment characteristics, and exposures common to the GW. GW-era veterans missing responses for the entire GWI symptom section, race, ethnicity, education, service branch, rank, or deployment status were excluded from the analyses (Figure 1). This study has been approved by the VA Central Institutional Review Board (CIRB) and the research oversight/ethics committees at each participating VA facility.

### 2.2. Supplementation of Data

Missing data were supplemented by the MVP Baseline Survey and VADIR. The source of VADIR data was from the Department of Defense (DoD) Defense Manpower Data Center and is used to determine eligibility for VA services. The following counts are the number of missing for each variable before supplementation: age (*n* = 626), sex (*n* = 152), race (*n* = 270), and ethnicity (*n* = 825), education (*n* = 61), and support of Operation Enduring Freedom (OEF) or Operation Iraqi Freedom (OIF) (*n* = 366). When available, missing variables were supplemented with data from EHRs and self-reported responses on the MVP Baseline Survey. Deployment information was supplemented with VADIR data (*n* = 212). Rank (*n* = 35,108, 97.8%) information was taken directly from VADIR data. When available, rank from August 1990 was used. If multiple ranks were listed during August 1990 then the rank associated with the earliest date was used (*n* = 76, 0.2%). If no rank was listed during August 1990, then the rank closest to but prior to August 1990 was used (*n* = 119, 0.3%). If that was also unavailable, then the first rank post August 1990 was used (*n* = 528, 1.5%). After data cleaning and quality control, complete information was available for 79.5% of respondents (*n* = 35,902) (Figure 1). Finally, unit component data was taken from VADIR data and service branch data was taken from the GW survey. Any remaining missing items could not be supplemented or data were not available for supplementation at the time of the data pull and subsequent analyses phase.

### 2.3. Measures

GWI phenotypes. GWI phenotypes used in this analysis were derived from the Kansas [8] and CDC [7] definitions [10,20,21]. Briefly, the Kansas and CDC definitions are based on veterans’ self-reported chronic symptoms. Survey participants were asked to report if each of a list of GWI symptoms was a persistent or a recurring problem and if so, its severity (mild, moderate, or severe). Participants were also asked a series of yes/no questions regarding 84 health conditions previously diagnosed by a healthcare provider. Using information on self-reported symptoms, symptom severity, and health conditions, we applied a SAS scoring algorithm as described previously [21] with slight modifications due to differences in question wording and available information (see Table A1 for details).

Kansas GWI phenotypes. We considered two phenotypes based on the Kansas definition [8]. The first phenotype assesses whether veterans met the Kansas symptom (KS Sym+) criteria of experiencing two or more mild symptoms or at least one moderate or severe symptom in at least three of six domains: fatigue/sleep problems, pain, neurologic/cognitive/mood, gastrointestinal, respiratory, and skin (Table A1).

The second Kansas GWI phenotype (KS Sym+/Dx−) indicates whether veterans met the KS Sym+ criteria and reported “No” to having been told by a doctor or healthcare provider they had any one of a series of exclusionary health conditions, such as a diagnosis of cancer (brain, breast, colon, lung, prostate, melanoma, other), diabetes, heart disease (heart attack, coronary artery disease, congestive heart failure), stroke (stroke and transient ischemic attack), infection (HIV, tuberculosis, hepatitis C), liver disease, lupus, mental health (schizophrenia and bipolar disorder), and neurological (multiple sclerosis and traumatic brain injury (TBI)) conditions. Veterans who met the exclusionary conditions were retained in the analysis as non-cases. Ambiguous responses of “Yes” and “No” by participants to a survey question were recoded as missing. 

Both phenotypes (KS Sym+ and KS Sym+/Dx−) are potentially valuable to explore. While the latter reflects the IOM recommended GWI criteria, the former focuses solely on symptoms reported by veterans. As we are unable to determine whether the symptoms are attributable to existing diagnosed conditions and/or other factors such as GWI, this study examined both phenotypes of KS Sym+ and KS Sym+/Dx−.

CDC GWI phenotypes. We considered two phenotypes based on the CDC definition [7]. The first one, CDC GWI, was met if veterans stated they had at least one symptom in two of the three symptom domains including fatigue, musculoskeletal, and mood/cognition (See Table A1). The second phenotype was CDC GWI severe which was met if veterans rated at least one symptom as “severe” in two or more symptom domains. Ambiguous responses of “Yes” and “No” by participants to a symptom question and multiple severity responses to the severity rating questions were recoded as missing. If respondents reported a symptom severity, responses of “missing”, “No”, “Yes and No” to the applicable symptom question were coded as “Yes”. 

Sociodemographic & healthcare utilization characteristics. Self-reported information from the 1990–1991 GW-era survey provided information on age at time of survey, sex (male or female), race (White, Black/African American, American Indian/Alaskan Native, Asian, other, multiple responses), ethnicity (Hispanic or Non-Hispanic), education (≤High School Diploma/GED, some college credit but no degree, Associates Degree, Bachelor’s Degree, ≥Master’s Degree), and reliance on VA for healthcare (all care, >half, ≤half, and none). Asian race includes Chinese, Japanese, Asian Indian, Other Asian, Filipino, and Pacific Islander. Hispanic includes Mexican, Puerto Rican, Cuban, or Other Spanish/Hispanic/or Latino.

Military characteristics. Military characteristic variables asked on the GW-era survey included military service branch during 1990–1991 (Army, Navy, Air Force, Marines Corps). Veterans reported if they deployed to the Gulf during the 1990–1991 GW-era (yes/no) and their dates of deployment. They also reported if they deployed in support of Operation Enduring Freedom (OEF) or Operation Iraqi Freedom (OIF) (yes/no), their unit component (Active Duty, National Guard, and Reserve), and their rank (Enlisted, Officer, and Warrant officer). 

### 2.4. Statistical Analyses

Descriptive statistics are presented, and chi-square tests and *t*-tests were used to compare demographic and military characteristics of deployed and non-deployed veterans. Associations between deployment status and (a) the four GWI phenotypes (Kansas Sym+, Kansas Sym+/Dx−, CDC GWI, CDC GWI Severe), and (b) the Kansas exclusionary conditions were evaluated using logistic regression in both unadjusted and adjusted models (adjusted for age [as a continuous variable in the model with odds ratios calculated for 10-year increase in age], sex, race, ethnicity, education, service branch, rank, and unit component). Age, race, ethnicity, education, service branch, rank, and unit component were used in the adjusted logistic models as they were found to be the most important variables after bivariate analyses, a review of the literature, and discussions with co-authors. Race and ethnicity were used in our model to address potential health and healthcare inequities stemming from social and economic inequities. Additionally, associations between exclusionary conditions and deployment status, were evaluated using logistic regression models, both unadjusted and adjusted (adjusted for the same covariates mentioned above). Associations between sociodemographic and military characteristics and GWI phenotypes were assessed using logistic regression in unadjusted and similarly adjusted models. A sensitivity analysis was conducted in which only complete cases were included in the analyses and any observations with missing data for covariates were dropped and all analyses re-run again. Statistical significance was set at *p* < 0.05 (two-sided). All analyses were conducted using SAS Enterprise Guide 8.2 software, SAS Institute Inc., Cary, NC, USA [22] and Figure 2 was generated using R Studio 2022.02.0 Build 443 software, Rstudio Team, Vienna, Austria [23].

## 3. Results

### 3.1. Characteristics of GW Era Veterans by Deployment Status

Among 109,976 eligible GW-era MVP participants who received GW surveys, 45,169 (41.1%) returned the survey; 13,107 (36.5%) were deployed to the Persian Gulf region in 1990–1991. Overall, the study sample was predominantly male (82.9%) and White (71.5%) with some notable differences between the deployed and non-deployed (Table 1). Deployed veterans (mean age 58.9 years, standard deviation [SD] 7.8 years) were younger than non-deployed (mean age 62.3 years, SD 8.3 years), included more self-reporting Black/African American race (deployed 22.1% vs. non-deployed 16.5% Black and deployed 66.6% vs. non-deployed 74.3% White), and comprised more male veterans (deployed 89.3% vs. non-deployed 79.2%). The most common service branch was the Army (deployed 52.1% vs. non-deployed 47.1%), with most respondents being on Active Duty (deployed 78.8% vs. non-deployed 66.1%) and in the enlisted ranks (deployed 82.0% vs. non-deployed 72.5%) during 1990–1991.

### 3.2. Associations of GWI Phenotypes by GW Deployment Status

KS Sym+ (deployed 77.9%; non-deployed 60.9%) and CDC GWI (deployed 87.9%; non-deployed 77.1%) phenotypes were identified in more than 60% of both deployed and nondeployed veterans. In contrast, both KS GWI (Sym+/Dx− deployed 24.8%; non-deployed 19.5%) and CDC-severe (deployed 27.0%; non-deployed 13.7%) were less prevalent (Table 2, Panel A). In both unadjusted and adjusted models, deployed veterans had higher odds of meeting criteria for each of the Kansas and CDC GWI phenotypes than non-deployed; the adjusted odds ratio (aOR) ranged from 1.25 to 2.15, all *p* < 0.05. 

The association between GWI and deployment was consistent when considering components of each of the definitions. For Kansas GWI symptom domains, the three most prevalent symptom domains were neurologic/cognitive/mood (81.9%), fatigue/sleep problems (73.9%), and pain (71.5%) (Table 2, Panel B). Compared to non-deployed, deployed had higher adjusted odds of meeting each of the Kansas GWI phenotypes with aORs ranging from 1.59 to 2.04 (all *p* < 0.05). Although the presence of one or more exclusionary conditions was very similar in the deployed (57.4%) and non-deployed (56.5%) groups, the deployed had slightly higher odds of having one or more exclusionary conditions (aOR = 1.12, 95% CI = 1.07, 1.18).

Likewise for the CDC GWI and CDC GWI severe components, both the unadjusted and adjusted models indicated that deployed veterans had higher adjusted odds of meeting each of the three CDC symptom domains compared to non-deployed (aORs ranging from 1.53 to 2.15, *p* < 0.05) (Table 2, Panel C).

### 3.3. Frequency and Associations of Exclusionary Conditions Deployment Status

Among all respondents, the most frequent exclusionary conditions (Table 3) were diabetes (25.4%), cancer (16.9%), heart disease (15.2%), neurological disorders (7.2%), and stroke (7.0%). Compared to non-deployed veterans, deployed veterans had higher adjusted odds for the following exclusionary conditions: cancer, stroke, liver disease, mental health, and neurological disorders (aORs ranging from 1.13 to 1.23, *p* < 0.05). The exclusionary conditions with the largest adjusted odds ratios include schizophrenia (aOR = 1.31), traumatic brain injury (aOR = 1.29), melanoma (aOR = 1.25), liver disease (aOR = 1.20), and stroke (aOR = 1.15).

### 3.4. Associations of GWI Phenotypes with Demographic Characteristics

We stratified our sample by deployed and non-deployed veterans and then examined associations between GWI and demographic characteristics within strata. Figure 2 and Table A3 provides an overview of this analysis demonstrating general patterns of association of demographic variables with the four GWI phenotypes of interest. Some patterns of association were very similar among deployed and non-deployed veterans. Table A2 (unadjusted analyses) and Figure 2 and Table A3 (adjusted analyses) provides detailed ORs, aORs, and 95% CIs. Older age was associated with lower odds of each GWI phenotype (aOR seen in both deployed and non-deployed range from 0.58 to 0.83, *p* < 0.05). Additionally, females (compared to males) had higher adjusted odds for each GWI phenotype regardless of deployment status (aORs ranging from 1.30 to 1.73, *p* < 0.05). 

**Figure 2 ijerph-20-00258-f002:**
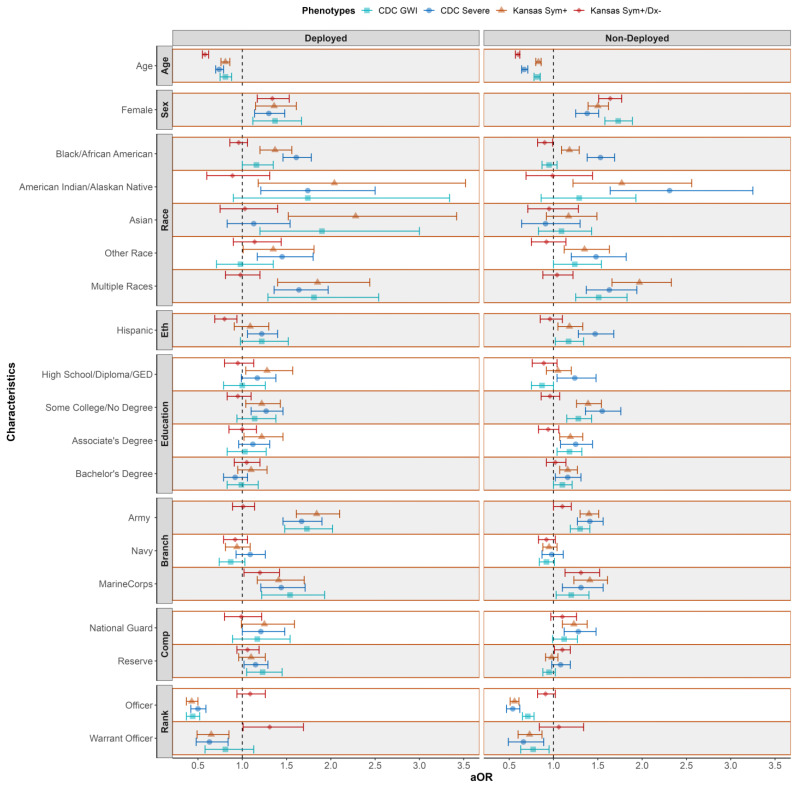
Adjusted Associations of Gulf War Illness Phenotypes with Demographic Characteristics Stratified by Deployment Status. Legend: Reference groups include male, white, non-Hispanic, ≥master’s degree, Air Force, Active Duty, and Enlisted. CDC = Centers for Disease Control and Prevention, GWI = Gulf War Illness, Kansas Sym+ = Kansas Symptoms, Kansas Sym+/Dx− = Kansas (symptoms and no exclusionary conditions), Eth = ethnicity, Branch = service branch, comp = unit component and aOR = adjusted odds ratio. Adjusted for age (as a continuous variable in the model with odds ratios calculated for 10-year increase in age), sex, race, ethnicity, education, service branch, rank, and unit component. Statistically significant if 95% confidence interval does not include 1.00 aORs are derived from eight models, two (deployed and non-deployed) for each GWI phenotype (KS Sym+, KS Sym+/Dx−, CDC GWI and CDC Severe GWI). Data are from the Department of Veterans Affairs’ (VA) Cooperative Studies Program 2006/Million Veteran Program (MVP) 029 cohort, Genomics of GWI (VA CSP 2006/MVP 029 Project) with a total study population size of 35,902; 13,107 deployed and 22,795 non-deployed.

Among the deployed, White veterans had lower adjusted odds of meeting the KS Sym+ phenotype than each of the other racial groups (*p* < 0.05) but did not differ from the other racial groups in their odds of meeting the KS GWI Sym+/Dx− phenotype. In contrast, non-Hispanic veterans did not differ from Hispanic veterans on the adjusted odds of meeting KS Sym+ but had lower adjusted odds than non-Hispanic veterans for meeting the KS GWI Sym+/Dx− phenotype; a similar pattern was observed for associations between racial and ethnic groups and GWI phenotypes among non-deployed veterans. A higher risk of GWI was found among those with some college but no degree for KS Sym+ and CDC Severe phenotypes among the deployed while a higher risk of GWI was found among non-deployed veterans with some college but no degree compared to those with a master’s degree or higher education for three of four GWI phenotypes (not Sym+/Dx−) (All *p* < 0.05).

For both deployed and non-deployed veterans, individuals who served in the Air Force had lower adjusted odds of meeting the KS Sym+, CDC GWI and CDC Severe GWI phenotypes compared to individuals who served in the Army or Marine Corps (*p* < 0.05). The exception was that Army did not differ from the Air Force for either deployed or non-deployed for the KS Sym+/Dx−. Among the deployed, individuals who served in the Reserve components had higher odds of CDC GWI and CDC severe GWI than veterans in the Active Duty components (*p* < 0.05).

In both the deployed and non-deployed samples, veterans who had served as officers had substantially lower adjusted odds for the KS Sym+, CDC GWI and CDC Severe GWI phenotypes (aORs ranging from 0.43 to 0.71) compared to veterans who had served in the enlisted ranks but not the KS Sym+/Dx− phenotype. Warrant officers also had lower adjusted odds of KS Sym+ and CDC GWI Severe compared to enlisted veterans (aORs ranging from 0.63 to 0.73; *p* < 0.05) regardless of deployment status. For the KS Sym+/Dx− phenotype among the deployed, warrant officers were at increased risk of GWI compared to enlisted (aOR 1.31; *p* < 0.05).

### 3.5. Missingness Analysis

For our analytic samples, we documented the number and percent of cases where item non-response prevented us from ascertaining a given phenotype (Table A4). For our total sample (N = 35,902), we lacked information for 5.7% of KS Sym+ phenotype, 11.0% of KS GWI Sym+/Dx−, 0.8% of CDC GWI phenotype, and 10.1% of CDC GWI Severe phenotypes. Both deployed and non-deployed had similar distribution of missingness percentages for these phenotypes. 

Complete case analyses were conducted for all variables found in Table 1, Table 2 and Table 3 and Figure 2 in the main analyses, which reduced the sample size to 20,194. In these analyses, we found similar results compared to the primary analyses with the full sample size (Table A5, Table A6, Table A7, Table A8 and Table A9).

## 4. Discussion

This study documented the prevalence of four GWI phenotypes among veterans in the VA CSP 2006/MVP 029 cohort, the largest sample of GW-era veterans available for research to date, thereby laying the foundation for future analyses to understand the epidemiology and biology of GWI. In this cohort, 67.1% met the KS Sym+ phenotype and when exclusionary health conditions were considered, 21.5% met the Kansas GWI case definition (KS Sym+/Dx−). Similarly, 81.1% of our cohort met the CDC GWI phenotype and when symptom severity was considered, 18.6% met the CDC GWI severe phenotype. Importantly, we consistently observed that deployed relative to non-deployed veterans had higher odds of meeting each GWI phenotype (aORs from 1.25 to 2.15, all *p* < 0.05). These findings are similar to the adjusted odds reported in the CSP 585 pilot study [10] except the pilot reported the highest adjusted odds ratio in the CDC GWI severe phenotype (aOR 2.67; 95% CI: 1.79–3.99) while our study observed that the highest adjusted odds ratio occurred in the KS Sym+ phenotype (aOR 2.15; 95% CI: 2.03–2.27). Similar to other more recent studies, the prevalence of symptoms and diagnosed health conditions reported by GW-era veterans in the CSP 2006 cohort were substantially higher than observed in studies conducted in the decade after the Gulf War, when both the CDC and Kansas case definitions were established [7,8].

Findings from our study revealed that older age was associated with lower odds of meeting the GWI phenotypes than younger age. Although this association has been reported in other studies of deployed GW veterans [6,10,24], in this current study, somewhat surprisingly, this finding held in both non-deployed and deployed samples--even after adjusting for military rank during the war. Previous explanations for the observed relationship between younger age and greater risk of GWI included the lower educational attainment of the younger servicemembers at the time of the deployment, their relative inexperience and sense of control during deployment reflected in their lower military rank, and their greater likelihood of being directly exposed to deployment-related toxins that may be associated with GWI. Indeed, most studies have shown GWI and related symptoms to be more common among enlisted personnel than officers [8,25]. We alternatively hypothesized that we would find higher GWI symptom rates among older veterans in our cohort due to the accumulation of chronic physical symptoms (e.g., joint pain, skin conditions) with increasing aging. This reflects a limitation of the current research case definitions of GWI; the lack of a required temporal association with the putative exposures during the 1990–1991 GW. However, our results did not show older age to be associated with increased risk of the symptoms based GWI phenotypes. These results warrant further study, especially as GW veterans age. Biomarkers of aging, such as epigenetic age acceleration, may be useful in untangling the relationship between age and GWI case status [26,27].

An important contribution of our study was that we separately examined the associations of demographic characteristics with four GWI phenotypes that differ substantially in prevalence. This approach revealed that women, regardless of deployment status, had higher odds of meeting the GWI phenotypes compared to men. Similarly, analysis of the Fort Devens Cohort reported higher odds of GWI among women compared to men (N = 945; aOR = 1.8, *p* < 0.05) [28]. Additionally, our study showed that while non-deployed Black and Hispanic veterans were more likely than White veterans to meet the KS Sym+, the opposite association was observed for the KS GWI (Sym+/Dx−) phenotype among Black veterans. A key difference between the two phenotypes is that while KS Sym+ does not consider exclusionary health conditions, KS GWI (Sym+/Dx−) excludes veterans who report certain health conditions and in the U.S. the accelerated accumulation of such health conditions along racial and ethnic lines is well-documented [29,30]. This association calls for careful consideration of how to account for underlying health conditions and their attribution to deployment and military exposures [31]. This call for careful consideration of how to account for health conditions is further reified by the concern that veterans of the Gulf War may be at increased risk for experiencing certain health conditions [32] and/or experiencing accelerated aging as partially measured by earlier onset of certain health conditions [33].

The phenotypes presented here will be used in VA CSP 2006/MVP 029 [20] to understand how genetic variation is associated with the GWI phenotypes and to identify potential pathophysiologic underpinnings of GWI, pleiotropy with other traits, and gene by environment interactions [20]. Initial results from the CSP 585 pilot study suggest that relationships between GWI, genes, and exposures to toxins may partially account for symptoms that veterans experience [34]. Recent analyses have successfully replicated earlier reports of the association of GWI with gene-by-toxicant interactions, specifically for the rs662 variant in the PON1 gene and exposure to the chemical weapon, Sarin [35]. However, such candidate gene studies fail to contextualize findings within genome-wide variation and may not capture interactions between military-related exposures and underlying genetic susceptibility. As the largest dataset of GW-era veterans, VA CSP 2006/MVP 029 will be able to perform genome-wide interaction studies with a broader range of GW exposures, allowing for in-depth comparison of exposures and underlying genetic susceptibility to GWI. These studies cannot be performed without the full description of the GWI phenotypes detailed here. 

### Strengths and Limitations

We acknowledge several limitations to our study including that only 41.1% of eligible MVP source population participated, which may have resulted in selection bias and reduced generalizability. MVP participants at the time of cohort creation were almost exclusively users of the Veterans Health Administration, introducing further sources of selection bias. Moreover, because the field of GWI research relies on case definitions based on self-reported symptoms without an objective diagnostic biomarker, we cannot rule out misclassification as a contributing bias. Furthermore, reporting bias could have occurred if veterans over- or underreported symptoms or medical conditions and if differences in reporting varied by deployment status. The use of race and ethnicity may be problematic as contemporary science has demonstrated that race is a social category with no basis in biology and instead analyses using race and ethnicity should be used as a proxy for unmeasured consequences of racism [36,37]. Given that two mild symptoms or one severe symptom are needed to attain a positive Kansas domain score, domains with more symptoms such as neurological symptoms may appear more common than other domains composed of fewer domains such as skin symptoms. Additionally, how symptom severity is incorporated into determining GWI case status may greatly affect how veterans are classified as evidenced by the difference seen here in the percent of veterans who met CDC GWI (81.1%) vs. CDC GWI Severe (18.6%). Therefore, future investigations should carefully consider how symptom severity is used for determining GWI case status.

The strengths of this study are the large sample size, which allowed for more representation of subpopulations of GW-era veterans, including by age, sex, race, ethnicity, education, and military service. Previous studies often did not have sufficient sample size and power to assess these differences in these demographic and military subgroups [38,39,40]. Additionally, we collected rich data from surveys, especially data on veterans’ self-reported symptoms, health conditions, and military service and GW deployment characteristics, which are incompletely and infrequently documented in medical records. Another important strength of the current study is that data were collected in 2018–2019, more than 27 years after the Gulf War. The study therefore provides an updated, detailed description of symptoms and conditions affecting GW-era veterans, decades after their return from service in the 1990–1991 Gulf War. 

## 5. Conclusions

Using data from GW-era veteran participants of the Million Veteran Program (VA CSP 2006/MVP 029), we confirmed prior reports of enduring differences in GWI between deployed and non-deployed GW-era veterans. The comparability of VA CSP 2006/MVP 029 characteristics with other GW-era veteran samples and the consistency of excess symptoms associated with GW deployment, coupled with the availability of genetic and exposure information, suggest that the VA CSP 2006/MVP 029 cohort and data resource offer a powerful tool for future inquiry into understanding the biological and environmental factors that are associated with GWI.

## Figures and Tables

**Figure 1 ijerph-20-00258-f001:**
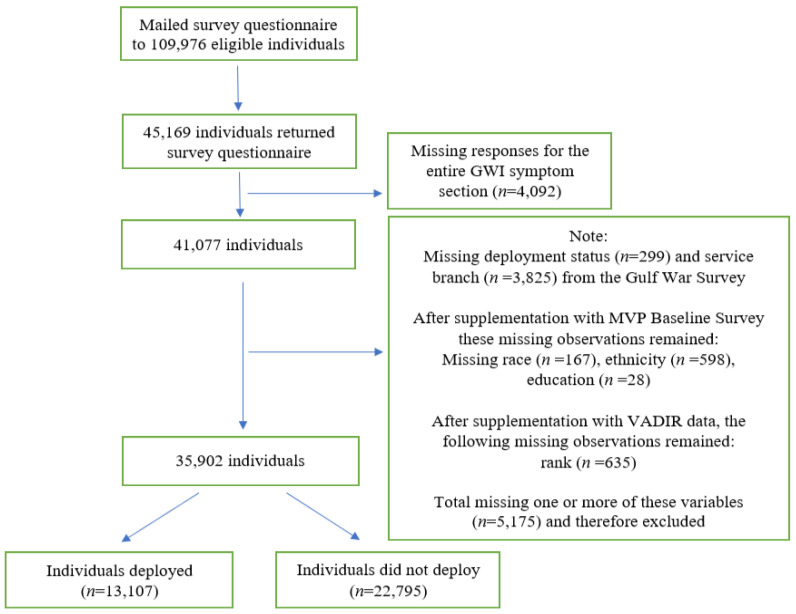
Flow chart for inclusion in the analytic dataset. Legend: Data are from the Department of Veterans Affairs’ (VA) Cooperative Studies Program 2006/Million Veteran Program (MVP) 029 cohort, Genomics of GWI (VA CSP 2006/MVP 029 Project). Supplemental data are from the MVP Baseline Survey and the VA Department of Defense Identity Repository (VADIR). The source of VADIR data was from the Department of Defense (DoD) Defense Manpower Data Center and data were provided by the VA. VADIR data are used to determine eligibility for VA services.

**Table 1 ijerph-20-00258-t001:** Characteristics of Gulf War Era Veterans in the VA CSP 2006/MVP 029 cohort by deployed and non-deployed status.

Characteristics	All (N = 35,902)	Deployed (n = 13,107)	Not Deployed (n = 22,795)
	n (%)	n (%)	n (%)
Age			
Mean (SD) Median	61.0 (8.3) 60.7	58.9 (7.8) 58.0	62.3 (8.3) 62.3
Min–Max	44.6–90.8	45.1–88.6	44.6–90.8
Age Group			
44–49	3467 (9.7)	1760 (13.4)	1707 (7.5)
50–59	13,419 (37.4)	5817 (44.4)	7602 (33.3)
60–69	13,190 (36.7)	4258 (32.5)	8932 (39.2)
70+	5826 (16.2)	1272 (9.7)	4554 (20.0)
Sex			
Male	29,770 (82.9)	11,711 (89.3)	18,059 (79.2)
Female	6132 (17.1)	1396 (10.7)	4736 (20.8)
Race			
White	25,659 (71.5)	8727 (66.6)	16,932 (74.3)
Black/African American	6653 (18.5)	2902 (22.1)	3751 (16.5)
American Indian/Alaskan Native	325 (0.9)	146 (1.1)	179 (0.8)
Asian ^1^	583 (1.6)	253 (1.9)	330 (1.4)
Other	1181 (3.3)	497 (3.8)	684 (3.0)
Multiple Responses	1501 (4.2)	582 (4.4)	919 (4.0)
Ethnicity			
Hispanic ^2^	3033 (8.4)	1283 (9.8)	1750 (7.7)
Non-Hispanic	32,869 (91.6)	11,824 (90.2)	21,045 (92.3)
Highest achieved education level			
≤High school diploma/GED	3185 (8.9)	1510 (11.5)	1675 (7.3)
Some college credit, but no degree	9535 (26.6)	3931 (30.0)	5604 (24.6)
Associates Degree	5701 (15.9)	2177 (16.6)	3524 (15.5)
Bachelor’s Degree	8075 (22.5)	2805 (21.4)	5270 (23.1)
≥Master’s degree	9406 (26.2)	2684 (20.5)	6722 (29.5)
Reliance on VHA health care in the last year			
All care	13,056 (36.4)	5154 (39.3)	7902 (34.7)
>half	10,398 (29.0)	3771 (28.8)	6627 (29.1)
≤half	9370 (26.1)	3155 (24.1)	6215 (27.3)
None	3014 (8.4)	1004 (7.7)	2010 (8.8)
Missing	64 (0.2)	23 (0.2)	41 (0.2)
Service branch			
Army	17,563 (48.9)	6825 (52.1)	10,738 (47.1)
Navy	7991 (22.3)	2981 (22.7)	5010 (22.0)
Air Force	7616 (21.2)	1930 (14.7)	5686 (24.9)
Marine Corps	2712 (7.6)	1363 (10.4)	1349 (5.9)
Deployed in support of OEF or OIF			
Yes	9829 (27.4)	4037 (30.8)	5792 (25.4)
No	26,001 (72.4)	9037 (68.9)	16,964 (74.4)
Missing	72 (0.2)	33 (0.3)	39 (0.2)
Rank			
Enlisted	27,285 (76.0)	10,754 (82.0)	16,531 (72.5)
Officer	7701 (21.4)	1984 (15.1)	5717 (25.1)
Warrant Officer	916 (2.6)	369 (2.8)	547 (2.4)
Unit Component			
Active Duty	25,395 (70.7)	10,332 (78.8)	15,063 (66.1)
National Guard	2518 (7.0)	626 (4.8)	1892 (8.3)
Reserve	7989 (22.3)	2149 (16.4)	5840 (25.6)

VA = Department of Veterans Affairs; CSP = Cooperative Studies Program; MVP = Million Veteran Program; SD = Standard Deviation; GED = General Educational Development Test; VHA = Veterans Health Administration; OEF = Operation Enduring Freedom; OIF = Operation Iraqi Freedom. ^1^ Asian includes Chinese, Japanese, Asian Indian, Other Asian, Filipino, and Pacific Islander. ^2^ Hispanic includes Mexican, Puerto Rican, Cuban, other Spanish/Hispanic/or Latino/or multiple responses. T-test for age as continuous variable by deployment status *p*-value ≤ 0.0001. All chi-square tests for categorical variables by deployment status *p*-value ≤ 0.0001.

**Table 2 ijerph-20-00258-t002:** Associations of Gulf War Illness Phenotypes with 1990–1991 Gulf War Deployment Status.

				Association with Deployment			
Measure	All (N = 35,902) %	Deployed (n = 13,107) %	Not Deployed (n = 22,795) %	OR	(95% CI)	aOR ^1^	(95% CI)
Panel A. GWI Case Status-Related Measures							
Kansas Symptoms (Sym+)	67.1	77.9	60.9	2.44	(2.31, 2.57)	2.15	(2.03, 2.27)
Kansas GWI (met symptoms; no exclusionary conditions [Sym+/Dx−])	21.5	24.8	19.5	1.38	(1.31, 1.45)	1.25	(1.18, 1.32)
CDC GWI	81.1	87.9	77.1	2.19	(2.06, 2.33)	1.99	(1.86, 2.12)
CDC GWI Severe	18.6	27.0	13.7	2.45	(2.31, 2.58)	2.06	(1.95, 2.19)
Panel B. Kansas GWI Components							
Symptom Domain							
Fatigue/sleep problems (4 symptoms)	73.9	83.4	68.4	2.37	(2.24, 2.51)	2.04	(1.93, 2.17)
Pain (3 symptoms)	71.5	78.4	67.5	1.81	(1.72, 1.91)	1.59	(1.50, 1.68)
Neurologic/Cognitive/Mood (14 symptoms)	81.9	88.8	77.9	2.30	(2.14, 2.46)	2.03	(1.89, 2.18)
Gastrointestinal (3 symptoms)	30.9	41.6	24.7	2.20	(2.10, 2.31)	1.98	(1.89, 2.08)
Respiratory (3 symptoms)	33.3	41.7	28.5	1.83	(1.75, 1.92)	1.74	(1.65, 1.82)
Skin (2 symptoms)	25.9	33.0	21.8	1.96	(1.87, 2.06)	1.88	(1.78, 1.98)
Exclusionary Conditions ^2^	56.8	57.4	56.5	1.08	(1.03, 1.13)	1.12	(1.07, 1.18)
Panel C. CDC GWI Components							
Symptom Domain							
Fatigue (1 symptom)	60.6	72.6	53.6	2.34	(2.23, 2.45)	2.15	(2.04, 2.26)
Musculoskeletal (3 symptoms)	86.9	90.2	85.1	1.65	(1.54, 1.77)	1.53	(1.42, 1.64)
Mood/Cognition (6 symptoms) ^3^	83.4	89.4	79.9	2.14	(2.00, 2.28)	1.92	(1.79, 2.05)
Severe Symptom Domain							
Fatigue (1 symptom)	11.2	16.7	8.0	2.33	(2.18, 2.49)	2.09	(1.94, 2.24)
Musculoskeletal (3 symptoms)	25.3	32.0	21.4	1.77	(1.68, 1.86)	1.53	(1.45, 1.61)
Mood/Cognition (6 symptoms) ^3^	28.5	39.0	22.5	2.36	(2.25, 2.48)	1.98	(1.87, 2.08)

CDC = Centers for Disease Control and Prevention; VA = Department of Veterans Affairs; CSP = Cooperative Studies Program; OR = Odds Ratio; aOR = Adjusted Odds Ratio; CI = confidence interval. ^1^ adjusted for age (as a continuous variable in the model with odds ratios calculated for 10-year increase in age), sex, race, ethnicity, education, service branch, rank, and unit component. Statistically significant if 95% confidence interval does not include 1.00. ^2^ Participants were asked if they were ever told by a doctor or healthcare that they had any of the following exclusionary conditions: cancer (brain, breast, colon, lung, prostate, melanoma, other), diabetes, heart disease (heart attack, coronary artery disease, congestive heart failure), stroke (stroke and transient ischemic attack), infection (HIV, tuberculosis, hepatitis C), liver disease, lupus, mental health (schizophrenia and bipolar disorder), and neurological (multiple sclerosis and traumatic brain injury (TBI)) conditions. ^3^ The six symptoms were based on seven items in the VA CSP 2006/MVP 029. Specifically, to reflect the original CDC wording, the highest level of symptom presence/severity from questions regarding “Difficulty remembering recent information” and “Difficulty concentrating” were combined to create one symptom category.

**Table 3 ijerph-20-00258-t003:** Frequency and Associations of Veteran-Reported Exclusionary Conditions by 1990–1991 Gulf War Deployment Status.

Condition	All (N = 35,902) (%)	Deployed (n = 13,107) (%)	Not Deployed (n = 22,795) (%)	OR	95% CI	aOR	95% CI
Any exclusionary condition	56.8	57.4	56.5	1.08	(1.03, 1.13)	1.12	(1.07, 1.18)
Cancer	16.9	15.3	17.8	0.86	(0.81, 0.91)	1.13	(1.06, 1.20)
Brain Cancer	0.2	0.3	0.2	1.22	(0.79, 1.88)	1.27	(0.80, 1.99)
Breast Cancer	1.2	0.8	1.4	0.54	(0.43, 0.67)	1.09	(0.86, 1.38)
Colon Cancer	1.0	0.9	1.0	0.90	(0.72, 1.13)	1.03	(0.82, 1.30)
Lung Cancer	0.6	0.5	0.6	0.80	(0.59, 1.07)	1.01	(0.74, 1.37)
Prostate Cancer	6.0	5.3	6.4	0.83	(0.75, 0.91)	1.06	(0.96, 1.17)
Melanoma	4.3	4.1	4.4	0.92	(0.83, 1.03)	1.25	(1.11, 1.40)
Other Cancer	5.9	5.2	6.2	0.86	(0.79, 0.95)	1.04	(0.94, 1.15)
Diabetes	25.4	25.8	25.2	1.03	(0.98, 1.09)	0.98	(0.93, 1.03)
Heart Disease	15.2	14.7	15.4	0.95	(0.89, 1.00)	1.07	(1.00, 1.14)
Heart Attack	7.2	7.2	7.2	1.00	(0.92, 1.09)	1.09	(0.99, 1.19)
Coronary Artery Disease	11.3	10.8	11.6	0.93	(0.87, 0.99)	1.07	(1.00, 1.15)
Congestive Heart Failure	3.8	3.7	3.9	0.96	(0.86, 1.07)	0.98	(0.87, 1.10)
Stroke	7.0	7.2	6.9	1.05	(0.96, 1.14)	1.15	(1.06, 1.26)
Stroke	4.6	4.8	4.5	1.08	(0.98, 1.20)	1.14	(1.02, 1.26)
Transient Ischemic Attack	3.9	3.9	3.8	1.02	(0.91, 1.14)	1.19	(1.06, 1.34)
Infectious Disease	5.6	6.2	5.3	1.18	(1.07, 1.29)	1.01	(0.92, 1.12)
HIV	0.9	0.9	0.9	1.00	(0.80, 1.25)	0.64	(0.51, 0.81)
Tuberculosis	2.8	3.3	2.6	1.30	(1.15, 1.48)	1.14	(1.00, 1.31)
Hepatitis C	2.2	2.3	2.1	1.06	(0.91, 1.22)	0.99	(0.85, 1.16)
Liver Disease	4.5	5.2	4.1	1.32	(1.19, 1.46)	1.20	(1.07, 1.33)
Lupus	0.8	0.8	0.8	1.05	(0.83, 1.33)	1.22	(0.95, 1.57)
Mental Health	4.9	5.8	4.4	1.37	(1.24, 1.51)	1.16	(1.04, 1.28)
Schizophrenia	1.0	1.3	0.7	1.79	(1.45, 2.21)	1.31	(1.05, 1.65)
Bipolar Disorder	4.4	5.1	4.0	1.32	(1.19, 1.46)	1.14	(1.02, 1.27)
Neurological	7.2	8.8	6.4	1.42	(1.31, 1.54)	1.23	(1.13, 1.34)
Multiple Sclerosis	1.0	1.0	1.0	0.99	(0.80, 1.23)	1.00	(0.79, 1.25)
Traumatic Brian Injury	6.3	7.9	5.4	1.51	(1.38, 1.64)	1.29	(1.18, 1.41)

OR = Bivariate Odds Ratio; CI = confidence interval; aOR = Adjusted Odds Ratio, adjusted for age (as a continuous variable in the model with odds ratios calculated for 10-year increase in age), sex, race, ethnicity, education, service branch, rank, and unit component. Statistically significant if 95% confidence interval does not include 1.00.

## Data Availability

Due to the nature of this research, participants of this study did not agree for their data to be shared publicly, so supporting data is not available.

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
