# Peer review of "Association of Gulf War Illness with Characteristics in Deployed vs. Non-Deployed Gulf War Era Veterans in the Cooperative Studies Program 2006/Million Veteran Program 029 Cohort: A Cross-Sectional Analysis"

_ijerph, 2022, doi:10.3390/ijerph20010258_

Round 1

Reviewer 1 Report

A continuing problem in the definition of GWI is the use of mild vs moderate severities. This limitation is most apparent for CDC CMI criteria where mild symptoms in 2 or 3 domains indicate GWI. The result is overlap with veterans with other conditions (exclusions) and reduced sensitivity and specificity. This limitation of CDC CMI should be referenced. The same concern of using mild for diagnosis under the Kansas criteria prevails. 

How many veterans met Kansas and CMI criteria is only moderate or severe were required for diagnosis? This stratification would help develop future diagnostic criteria for GWI.

There is an intrinsic bias in the Kansas domains because each has a different number of complaints. Neuro has 14, and GI has 3. Given that 2 mild symptoms are needed to attain a positive domain score, it means there neurological symptoms will be more prevalent in Kansas+ groups.  Please comment in study limitations.

Table 2 gives the rates of exclusion criteria in deployed and nondeployed. Please provide this data in Panel A and C as well as Panel B for Kansas.

Table 3. Highlight the most significant associated conditions in the text: schizophrenia ORadj 1.31, TBI 1.29, melanoma 1.25, liver disease 1.20, and stroke 1.15. These are valuable for veterans, clinicians and the VA to be aware of and potentially focus screening and treatment efforts. 

What is the rate of sleep apnea in your population? 

Line 404: It seems veterans had more symptoms in the 2006 survey than ones in the 1990's. How do you account for the difference? Late onset of symptoms consistent with persistent neurotoxicity? Ageing is not responsible based on the parallel nondeployed group. Were there shortcomings in the types of questions asked or the types of groups interrogated such as hospital populations, and veterans who were deployed to the Gulf after June 1991 and did not have the full exposure history but were listed as GWI?

The exclusionary conditions have relevance to the criteria the VA uses for disability determinations of GWI and GWI veterans. That branch of the VA may learn from your work.

One of the purposes of scientific research into the health of veterans is to provide actionable information for the Veterans Administration to use in making proper diagnoses and disability determinations in veterans. How does your work apply to this scientific duty? VA does not appear to follow standard criteria for GWI such as the Kansas and CMI criteria used in this manuscript. Instead, they have adapted a version of the 1994 Fukuda CFS criteria, and have tried without success to apply the 2015 Institute of Medicine Systemic Exercise Intolerance Disease (SEID) criteria but without any data from VA epidemiology studies to support their change. I think it would be meritorious to comment on the controversy of diagnosis in the VA as a part of this manuscript. The outcome may be a movement towards better recognition and care of GWI veterans.

I include this excerpt from Federal Register to show why leadership by the scientific community and this manuscript are required for the benefit of veterans.

https://www.federalregister.gov/documents/2019/06/18/2019-12682/schedule-for-rating-disabilities-infectious-diseases-immune-disorders-and-nutritional-deficiencies

6354 Chronic fatigue syndrome (CFS):

Debilitating fatigue, cognitive impairments (such as inability to concentrate, forgetfulness, or confusion), or a combination of other signs and symptoms:

Which are nearly constant and so severe as to restrict routine daily activities almost completely and which may occasionally preclude self-care   100

Which are nearly constant and restrict routine daily activities to less than 50 percent of the pre-illness level; or which wax and wane, resulting in periods of incapacitation of at least six weeks total duration per year    60

Which are nearly constant and restrict routine daily activities from 50 to 75 percent of the pre-illness level; or which wax and wane, resulting in periods of incapacitation of at least four but less than six weeks total duration per year         40

Which are nearly constant and restrict routine daily activities by less than 25 percent of the pre-illness level; or which wax and wane, resulting in periods of incapacitation of at least two but less than four weeks total duration per year         20

Which wax and wane but result in periods of incapacitation of at least one but less than two weeks total duration per year; or symptoms controlled by continuous medication          10

Note: For the purpose of evaluating this disability, incapacitation exists only when a licensed physician prescribes bed rest and treatment.

Multiple commenters including individual veterans, Veterans Advocacy Organizations, Veterans Service Organizations, and other professional organizations expressed a wide range of concerns regarding the proposed changes to the definition of chronic fatigue syndrome (CFS) under § 4.88a and the name change for DC 6354. Commenters thought the name change of Chronic Fatigue Syndrome (CFS) to Systemic Exertion Intolerance Disease/Chronic Fatigue Syndrome (SEID/CFS) was unwarranted and that it would create unnecessary confusion among medical providers, including non-VA medical providers. Commenters also stated that that the new name, Systemic Exertion Intolerance Disease (SEID), has not been adopted by any federal agency, nor by researchers and clinicians and that the CDC, National Institutes of Health (NIH), research publications, and materials for patients and health care providers all use the term ME/CFS. Commenters felt that VA's use of the term SEID/CFS would introduce confusion among medical providers and patients at VA and reduce VA's ability to coordinate with other federal agencies.

Commenters expressed that the proposed changes to the definition of CFS does not conform to the Kansas Criteria (2000), the Centers for Disease Control (CDC) Chronic Multisymptom Illness (CMI) criteria, and to those used in VA-funded research into Gulf War Illness (GWI) and that the proposed definition is not compatible with the department of Defense (DoD) Congressionally Directed Medical Research Programs (CDMRP) for CMI. Commenters stated that VA's proposed combination of the Institute of Medicine (IOM) reevaluation of CFS as SEID with the 1994 Fukuda criteria for CFS presents an amalgamation that is not based in evidence nor discussed in any publications. The commenters expressed concern that VA did not follow any recommendations from the IOM, the Gulf War Research Advisory Committee (RAC), CDC, or other agencies and this combination is for an entirely new entity that is not known by World Health Organization, International Classification of Diseases, Tenth Revision (ICD-10) or other medical classification system and that the VA proposed definition is not compatible with the one mandated by DoD's CDMRP for CMI and the Kansas Criteria to qualify for GWI research funding.

Commenters noted that VA did not consult the RAC on these proposed changes and stated that the RAC is responsible for understanding the definitions and entirety of the condition. Commenters also were concerned that the proposed changes would leave those Gulf War veterans who receive care and services for CFS, vulnerable to VA manipulation of their care and services. The commenters suggested that CFS should be studied by the Gulf War research community, the veteran community, CFS researchers, the RAC, and independent medical professionals and that VA rely on the recommendations from these parties as a guide for new criteria updates and to ascertain if these changes are even warranted. Commenters also stated that VA would be directly and negatively impacting more than 300,000 Gulf War veterans suffering from Gulf War Illness by not relying on the studies from these parties and by combining, in whole or in part, the 2015 Systemic Exertion Intolerance Disease (SEID) and the 1994 Fukuda CDC criteria for Chronic Fatigue Syndrome (CFS) into what would be called SEID/CFS.

Commenters felt that VA's adoption of the Fukuda criteria is a step backwards that will perpetuate diagnostic inaccuracy and cause harm to Myalgic Encephalomyelitis/Chronic Fatigue Syndrome (ME/CFS) patients served by the VA. Commenters referenced the 2015 IOM Report to state that the Fukuda criteria were overly broad because they do not require the hallmark symptom of post-exertional malaise and should not be used because of the possibility of misdiagnosing patients with other conditions. Commenters believed that VA's reliance on outdated Fukuda diagnostic criteria would cause harm to veterans with ME/CFS through misdiagnosis and cause a mismatch with the diagnostic criteria in use elsewhere. Commenters suggested that VA adopt ME/CFS or ME/CFS/SEID title for the illness to stay in alignment with the greater ME/CFS community, to include patients, doctors, and researchers. Commenters felt that VA's proposed revisions were based on financial reasons in order to revoke benefits from existing veterans and prevent other veterans from receiving this combined diagnosis of SEID/CFS.

Commenters also provided questions and recommended that VA adopt ME/CFS instead of SEID/CFS; reject the Fukuda criteria; and adopt the IOM diagnostic criteria.

Another recommendation was for VA to revise § 4.88a to more closely mirror the diagnostic standard endorsed by the IOM and CDC and eliminate the listed exclusions to allow the veterans' examining and/or treating physician to make a final determination as to the appropriate diagnosis for veterans. In addition, commenters recommended that VA should broaden the group of medical professionals authorized to prescribe bed rest and treatment to meet the incapacitation standard.

Reviewer 2 Report

This is an important epidemiologic study that uses data from GW-era subjects of the Million Veteran Program (VA CSP 2006/MVP 029) cohort and provides updated data on GWI symptoms, conditions, and deployment status collected 27 years after GW exposures.  This is a large cohort and offers the advantage of associated genetic and exposure information collected from these participants.  In this manner this study is critical to providing the foundation for future investigation of the biological and environmental factors associated with GWI. The authors should be commended on a well-written manuscript with informative tables, text, and discussion of relevant data. 

Reviewer 3 Report

This article required an impressive amount of team work in order to further the ongoing work of the Department of Veterans Affairs’ Million Veteran Program, investigating effects of genes, lifestyle, and military exposure, and the health of GW era veterans. It accessed the largest sample of GW-era veterans available for research to date to conduct a contemporary cross-sectional analysis of the association of Gulf War Illness with demographic and military characteristics in deployed vs. non-deployed Gulf War era veterans and report the prevalence among GW-era MVP participants of four GWI phenotypes derived from Kansas and CDC definitions. These findings are meant to form the foundation for additional epidemiologic, clinical, and genetic analyses of this cohort and has value due to large sample size and its contemporary detailed descriptions of symptoms and conditions affecting GW era veterans.

Specific comments:

Overall the manuscript is sound, well organized and well written. Only the introduction was more difficult and confusing to read. The authors describe four phenotypes: Kansas non case, Kansas case, CDC case and CDC severe. For the two Kansas phenotypes they use the acronyms/terms "KS Sym+“and “KS Sym+/Dx-" both in the abstract  (line 51, 52) and the introduction, (line 78, 79, 80) without immediate explanation other than referring to earlier papers, Gifford 2021, ref [10], which paper does not anywhere use these terms/acronyms and then Gifford 2022 ref [18] where these terms are introduced and are more clearly explained in regards to reference to cases and non cases. While many readers may be already familiar with the earlier references, other less familiar readers would have to wade through these 2 references to clarify these acronyms. It would be helpful to expand the way these terms are defined within this paper immediately rather than just referring to earlier papers. The terms do go on to be explained better in section 2.3 Measures, (lines 188-201). Even here it would be helpful to clarify or explain how the phenotype, KS Sym+ would still be of value, as it meets criteria for GWI per symptoms but due to exclusions is phenotypically a non case, an apparent contradiction, and include and even repeat the earlier discussions of why this is an acknowledged ongoing question as was well stated by Gifford 2021 [10], “Efforts to develop a single, robust GWI case definition must address challenges related to characterizing an updated symptom profile uniquely associated with Gulf War service. To account for change over time, this requires studying symptoms which were reported in recent years. Research may investigate refining the definition as a function of which symptoms at what severity levels to include. Updated guidelines for defining GWI and applying case definitions for subgroups with and without comorbidities that might otherwise be considered an exclusionary condition could help researchers implement a unified approach for studying this condition.”

And as Haley et al put it, ref [35] “as the veterans aged, the exclusions eliminated too many valid cases”.

While the authors recognize these important limitation of the current research case definitions of GWI they do not advance a solution, nor do they repeat the important directions posited from the 2021 paper which could be re stated. Many other exposures and events, as well timing of onset of exclusions, in the intervening years can introduce significant confounding,

The limitations are correct to identify that 41% of eligible enrollees (109,000) agreeing to complete the questionnaire could limit generalizability.

Minor

The reference list appears correct, except an editing point, that many but not all references give only the first author, et al, which would be non-standard up to 6 co-authors.
